# Graphene Modification by Curcuminoids as an Effective Method to Improve the Dispersion and Stability of PVC/Graphene Nanocomposites

**DOI:** 10.3390/molecules28083383

**Published:** 2023-04-11

**Authors:** Sławomir Wilczewski, Katarzyna Skórczewska, Jolanta Tomaszewska, Magdalena Osial, Agnieszka Dąbrowska, Kostiantyn Nikiforow, Piotr Jenczyk, Hubert Grzywacz

**Affiliations:** 1Faculty of Chemical Technology and Engineering, Bydgoszcz University of Science and Technology, Seminaryjna 3 Street, 85-326 Bydgoszcz, Poland; jolanta.tomaszewska@pbs.edu.pl; 2Institute of Fundamental Technological Research, Polish Academy of Sciences, Pawińskiego 5B Street, 02-106 Warsaw, Poland; mosial@ippt.pan.pl (M.O.);; 3Faculty of Chemistry, University of Warsaw, Pasteura 1 Street, 02-093 Warsaw, Poland; 4Biological and Chemical Research Centre, University of Warsaw, Żwirki i Wigury 101, 02-089 Warsaw, Poland; 5Institute of Physical Chemistry, Polish Academy of Sciences, Kasprzaka 44/52, 01-224 Warsaw, Poland

**Keywords:** graphene, curcuminoids, poly(vinyl chloride), nanocomposites stability, polymer films

## Abstract

A large amount of graphene-related research is its use as a filler for polymer composites, including thin nanocomposite films. However, its use is limited by the need for large-scale methods to obtain high–quality filler, as well as its poor dispersion in the polymer matrix. This work presents polymer thin-film composites based on poly(vinyl chloride) (PVC) and graphene, whose surfaces were modified by curcuminoids. TGA, UV–vis, Raman spectroscopy, XPS, TEM, and SEM methods have confirmed the effectiveness of the graphene modification due to π–π interactions. The dispersion of graphene in the PVC solution was investigated by the turbidimetric method. SEM, AFM, and Raman spectroscopy methods evaluated the thin-film composite’s structure. The research showed significant improvements in terms of graphene’s dispersion (in solutions and PVC composites) following the application of curcuminoids. The best results were obtained for materials modified with compounds obtained from the extraction of the rhizome of *Curcuma longa* L. Modification of the graphene’s surface with these compounds also increased the thermal and chemical stability of PVC/graphene nanocomposites.

## 1. Introduction

Research concerning the application of graphene materials is carried out in many areas, such as medicine, chemistry, materials engineering, energetics, and electronics. A large area of graphene–related research regards its use as a filler for polymer nanocomposites [1,2,3,4,5]. Graphene materials (graphene, multilayer graphene, graphene oxide) show a considerable variation regarding their dispersibility in the polymer matrix, depending on the number of defects and layers and the presence of oxygen–containing functional groups. Furthermore, studies have shown that adding graphene to the polymer matrix can positively change the physicochemical properties of plastics [6,7,8,9,10].

An interesting and still relatively unexplored group of polymer nanocomposites with graphene are poly(vinyl chloride)–based materials (PVC/GN). Analysis of the literature has shown that in these materials with relatively well-dispersed graphene, regardless of the method of preparation (melt mixing, solvent casting, in situ polymerization method), result in an increase in tensile strength and modulus of elasticity [11,12,13,14,15,16], as well as in increased impact strength [16,17]. Such materials are also characterized by higher thermal stability [13,18,19], swelling agent resistance [20,21,22] and glass transition temperature (T_g_) [13,14,15,23,24], lower volume and surface resistivity compared to matrix material [14,25,26]. However, the improvement of poly(vinyl chloride) properties using graphene has limitations, mainly due to the poor dispersibility of the filler in the matrix and insufficient interactions at the filler–polymer interface. Depending on the method of filler introduction, maximum properties are obtained in materials containing between 0.1 and 0.5 wt.% graphene. Filler agglomeration affects the deterioration of all the properties mentioned above of PVC/GN polymer nanocomposites.

The agglomerative behavior of graphene is due to strong π–π interactions and van der Waals forces [27,28,29,30]. The main preventive methods are based on covalent modification of chemical reactions with graphene’s carbon atoms [2,27,29] and non–covalent modification based on the adsorption of particles onto the material’s surface. Due to the functionalization mechanism, non-covalent methods to counteract graphene aggregation are divided into four main groups. The first includes methods related to π–π interactions [28,31,32,33,34]. The second stabilization mechanism is based on cation–π interactions [35,36]. Other methods apply hydrophilic–hydrophobic effects [2,27,29,37] and surface modifications of graphene with other nanoparticles [2,38,39]. Improving graphene’s dispersion in the PVC matrix is also based on the above methods. However, the covalent solutions presented so far [40,41,42,43,44] use reagents harmful to health and the environment, such as aniline, N–methylpyrrolidone, hydrazine, and terephthalaldehyde. Non–covalent methods, on the other hand, are mainly based on the use of plasticizers [14,25], which alter the properties of poly(vinyl chloride). Soft varieties of PVC in terms of properties can be considered a different group of materials compared to rigid PVC, so they also have entirely different practical applications [45].

The π–π interactions refer to non–covalent interactions (overlapping orbitals) between the π bonds of aromatic rings. However, this is a misleading description, because the direct sandwich overlap is electrostatically repulsive. Instead, we are dealing with staggered stacking (parallel displaced) or π–teeing (perpendicular T–shaped), which makes the described interactions electrostatically attractive. The displacement occurs due to the superposition of carbon atoms with partially negative charges over hydrogen atoms with partially positive charges. Similarly, T–type perpendicular overlap is electrostatically attractive because here, too, there is an overlap of different charges [32,33,46].

A promising method of graphene modification using π–π stacking is the modification of its surface with curcuminoids. The interaction between the aromatic rings of curcumin and the GN surface was used to obtain graphene by reducing graphene oxide, through ultrasonic exfoliation of graphite [31,34,47]. However, the most common use of the graphene–curcumin system is in drug delivery systems [48,49]. Curcuminoids, thanks to their high chemical and thermal stability (in combination with graphene, even up to 220 °C), are also attractive for applications in the modification of polymers, including PVC [50,51,52].

Therefore, the present work aimed to modify surface graphene with curcuminoids and to investigate the potential of this modification to improve graphene dispersion in poly(vinyl chloride)/graphene nanocomposites obtained by the solvent-casting method. In this work, we used a commercially available mixture of three types of curcumin (CU) (curcumin, demethoxycurcumin, and bisdemethoxycurcumin) and an extract of *Curcuma longa* L. (CE) obtained by extraction of the powdered rhizome of the plant, which was presented in our earlier work [22]. In addition to the aforementioned main ingredients, CE also contained Ar—curcumene, (-)—zingiberene, β—sesquiphellandrene, Ar—turmerone, α—turmerone, β—turmerone, (6R, 7R)—bisabolene and (E)—atlantone.

## 2. Results and Discussion

### 2.1. UV-Vis Spectroscopy

Before the morphology studies were conducted, the UV-vis studies were performed, in order to determine if the GN was successfully modified with curcuminoids. Figure 1 shows the UV–Vis spectra of graphene, curcumin, *Curcuma longa* L. extract, and GN–CU and GN–CE colloidal suspensions. The analysis showed that the maximum absorption for graphene was observed at 272 nm, corresponding to π–π* transitions of C=C bonds [53,54]. Two absorption bands were observed in the spectrum of curcumin; at 263 nm, with π–π* and C=C bonds, and at 422 nm, which corresponds to n–π* transitions of C=O bonds [54,55]. In the spectrum of the *Curcuma* extract, the described transitions were observed at 233 nm and 422 nm, respectively. The shift of the absorption maximum (233 nm) toward the ultraviolet spectrum is due to the presence of the other CE components. A maximum wavelength of 233 nm is characteristic of ar–turmerone [56,57]. Two absorption bands were observed on the GN–CU and GN–CE spectra at 269 nm and 414 nm, respectively, which were attributed to the π–π* transitions of C=C bonds and n–π* bonds of C=O. The presence of a band at 414 nm, characteristic of curcumin, indicates the effective functionalization of graphene [54,58]. In addition, the change in band position at 269 nm confirms the interaction of π–π graphene and curcuminoid–containing modifiers [21,55,59].

### 2.2. XPS Analysis

Next, X–ray photoelectron spectroscopy (XPS) was used to characterize graphene surfaces’ functionalization accurately. Figure 2 shows the XPS spectra of GN and curcuminoid–modified materials. In the image, one can see an increase in the intensity of the O1s band in materials after modification. It indicates the effective deposition of curcuminoids (which contain oxygen functional groups) onto the graphene surface [4,47]. Finally, the deconvolution of O1s and C1s peaks and the XPS characterization of GN, GN–CU, and GN–CE sheets are presented in Figure 3 and Table 1. Since graphene materials were modified, mainly sp^2^ hybridization was expected. Unfortunately, it was impossible to match sp^2^ in any way, so an additional sp^3^ hybridization peak was used. This fact may be due to the partial defecting of the graphene used and its partial oxidation (according to the manufacturer, about 2.5% was oxidized) [31,47,60].

The distance between the C sp^2^ and C sp^3^ peaks at 0.8 eV, especially for the starting sample, agrees very well with literature data on peak distances of thin carbon films, which are, respectively, 0.7 eV for graphene oxide and reduced graphene oxide [61], 0.8 eV for diamond-like carbon films [62,63], and 0.9 eV for graphite and graphite–like materials [64,65,66]. Furthermore, the position of the C sp^2^ peak around 284.5 eV is also characteristic of this type of material [64] and generally corresponds to values found in the literature (284.25–285 eV), including graphene materials [4,31,61,67].

The rest of the smaller carbon peaks (Figure 3, Table 1) of the carbon functional groups are characterized by the expected distance from the main peak C sp^2^ [68]. Considering the literature data, the chemical compounds of turmeric are composed of carbon–oxygen groups [69,70,71]. The obtained spectra for the samples with CU and CE confirm an increase in the proportion of oxygen groups (O1s), which was 0.6 at.% for GN and 5.3 and 3.3 at.% for GN–CU and GN–CE, respectively, which confirms the deposition of curcuminoids on the graphene’s surface. However, the changes in the π electrons (Table 1 C1s spectra) indicate the physical nature of their interactions as a result of non-covalent π–π stacking modification [63,72].

### 2.3. Thermogravimetric Analysis (TGA)

A thermogravimetric analysis was performed to determine the modifier’s content on the graphene surface. The test was carried out in the temperature range from 30 to 700 °C. The analysis results (Figure 4) showed that the graphene was thermally stable over the entire temperature range, in good agreement with the literature data [34,73]. The extract from *Curcuma longa* L was stable up to about 175 °C, and its degradation occurred between 175 °C and about 450 °C (Figure 4A) while the curcumin was stable up to 250 °C, and its decomposition occurred at temperatures ranging from 250 to 550 °C (Figure 4B). The differences in the thermal stability of these modifiers are due to the presence of additional components in the extract. Their decomposition, up to a temperature of about 320 °C, is related to the degradation of the –OH and –OCH_3_ groups. Furthermore, the curcuminoids are decomposed [74,75,76].

Both GN–CU and GN–CE showed thermal stability up to about 340 °C, and their decomposition was associated with the degradation of the organic modifier [34,73]. It is worth noting that the thermal stability of the modified graphene is well above the processing temperature of PVC (below 200 °C) [77,78,79]. This indicates that the proposed modification can also be used to obtain poly(vinyl chloride)/graphene nanocomposites by the melt mixing method. From the mass differences at 700 °C, the modifier’s content on the GN surface was determined by comparing the reference GN sample with the mass loss for GN–CU and GN–CE, respectively. The calculations showed that the mass of curcumin on the graphene surface was 11.4% (Figure 4B), while the mass of the extract was 6.9% (Figure 4A).

### 2.4. Morphology Characterization

The morphology of the materials was observed by transmission electron microscopy (Figure 5A–C) and scanning electron microscopy (Figure 5(A1–C1)). The observations showed that the modified graphene was a material composed of several layers, which was characterized by a crumpled morphology. This is typical of types of graphene with several layers and relatively large, flake diameters (according to the manufacturer, three-layered graphene with a diameter of up to 10 µm) [1,60,80].

Spherical curcuminoid particles (brighter white dots) can be seen in the SEM images (Figure 5(B1,C1)) of the modified graphene [81,82]. Simultaneously, they have a smaller diameter than those in GN–CU (Figure 5(B1)). On the other hand, in TEM images, it can be seen that both modifiers, CU and CE, on the graphene’s surface have a floral–spherical shape (Figure 5B,C) [83,84]. As in the SEM images, the curcuminoids from the extract (Figure 5C) were much better deposited on the graphene surface. They are well-dispersed, as present in all SEM pictures of the specimen, and have a smaller diameter. Figure 5(A2–C2) shows the uniform distribution of the flakes in the whole volume as a survey image. Based on the SEM images, it can be argued that CU and CE mainly cover the edges of the graphene flakes, but the TEM images, especially that of GN–CE, indicate that the modifier can cover the surface of graphene to a greater extent.

### 2.5. Raman Spectroscopy

The following step was the evaluation of the structure of graphene materials using Raman spectroscopy, an effective method for studying carbon materials [85]. Recorded spectra presented in Figure 6 for GN, GN–CU and GN–CE reveal the presence of the D-band, which is characteristic of graphene with a disordered carbon structure, and the G-band, typical for graphite materials, which, in the case of graphene, is also associated with the number of layers [5,86]. To estimate the size of graphene domains, the D- and G-band intensity (ID/IG) ratio was used, where this quantity is inversely proportional to the size of the crystallites [15,27,86,87]. In our materials, it was 1.5–1.6 (depending on the background treatment) for all fillers, indicating a high level of defects in curcuminoid-modified materials, which can increase their wettability by the polymer matrix. Furthermore, the intensity ratio 2D: G < 1 confirms the presence of multilayered, and not structuralized, material. The 2D band is broad and weak, and is slightly shifted towards higher frequencies (~2700 cm^−1^), which indicates the residual compression stress in the material. The RBS parameter (Raman Band Separtion) is relatively high and equals 287 (GN–CE) to 293 (GN–CU), whereas non modified filler has a value of ~291. This confirms the different mechanisms of carbon structure modification by both curcuminoids.

In paper [31], it was shown that the stacking of more graphene layers causes a shift of the G-band peak towards higher frequencies, and the Rs for the monolayer is generally higher than for a bilayer. In curcuminoid-modified materials, a shift toward higher wavenumbers was observed, from 1598 cm^−1^ for GN to 1600 cm^−1^ for graphene modified with curcumin and *Curcuma longa* L. extract, further confirming the better foliation of the material after surface functionalization with curcuminoids. Bands typical for disordered carbons are not visible (360 cm^−1^, 1620 cm^−1^).

### 2.6. Dispersion Stability Analysis

As we pointed out in our earlier work [21], in nanocomposites obtained by solvent casting, the critical step in determining the homogeneity of graphene dispersion in the PVC matrix is obtaining a stable filler dispersion in the polymer solution. Therefore, the stability of GN, GN–CU, and GN–CE dispersions in a 3% poly(vinyl chloride) solution was evaluated, from which thin polymer films were obtained after THF evaporation. The test was carried out using the multiple light-scattering method (used in the Turbiscan Lab Toulouse, France). Figure 7 shows the change in the percentage of backscattered and transmitted light as a function of the height of the measuring cell over 24 h. This time is sufficient to obtain a polymer film from a PVC solution at room temperature.

The analysis shows that the destabilization and stratification of graphene dispersion occur due to re-agglomeration and sedimentation of the filler [88,89]. The study demonstrated that the unmodified graphene does not form stable dispersions in PVC solution (Figure 7(A1,A2)), as evidenced by changes in transmitted and backscattered light [88,90]. Functionalization of the GN’s surface with curcuminoids improved its stability in PVC solution. Neither GN–CU nor GN–CE re-agglomerated, as evidenced by the absence of significant changes in backscattered light (Figure 7(B2,C2)) [89]. However, in Figure 7(B1,C1), we can see an increase in the solution’s transparency after about 16 h, especially for GN-CU. This is due to the gravitational descent of the filler particles.

The Turbiscan Stability Index (TSI) (a specific Turbiscan parameter) was determined to compare the destabilization kinetics of the prepared systems, in particular, the differences between the CU and CE effect. TSI is an efficient method of comparing several systems, where the higher its value is, the less stable the system is [91,92]. Other sources [21,28] distinguish a more precise division of TSI values, indicating that the dispersion is stable when TSI is less than five. When its value is between 5 and 20, we are dealing with a partially stable system; above 20, the dispersion is sedimentary. The results presented in Figure 7(A2–C2) show that the GN dispersion is stable for only 4 h, while modifying its surface resulted in extending this time to 24 h. TSI analysis proved that the most stable systems are formed by GN–CE, for which the discussed value, even after 24 h, was below 0.05.

### 2.7. Graphene Dispersion in Nanocomposites

The structure of nanocomposite polymer thin films obtained by solvent casting was studied using Raman spectroscopy and atomic force microscopy. In addition, the morphologies of the obtained materials were also observed on a macro- and micrometer scale, using images from digital photography and SEM. As mentioned in the introduction, the problem with graphene aggregation occurs in materials containing about 0.5 wt.% filler or more, so the structure evaluation was carried out for materials with 1 wt.% GN.

Regarding the Raman spectra of the nanocomposites (Figure 8), D– and G–bands typical of graphene and peaks originating from poly(vinyl chloride) were observed. Signals were assigned to the following chemical groups: a band at about 360 cm^−1^ to C–Cl in the trans configuration of the polymer [93], bands at 635 and 695 cm^−1^ to C–Cl stretching [93,94], a band at approximately 1113 cm^−1^ to C–C stretching [93], a band at ~1430 and 1498 cm^−1^ to C–H symmetrical stretching in CH_2_ group [21] and a band at approximately 2916 cm^−1^ to C–H asymmetrical stretching in CH_2_ group [21,93]. In addition, a peak was observed in PVC/1% GN–CU and PVC/1% GN-CE at about 1185 cm^−1^. It was attributed to the C–O bonds of curcuminoids [95]. The study did not reveal any new chemical bonds in PVC, which indicates the physical nature of filler–polymer interactions. Analysis of the graphene–derived bands showed changes in the fillers’ structures, due to their dispersion in the poly(vinyl chloride) matrix. Firstly, the RBS decreased for all materials, with 243 (GN–CE), 247 GN, and 249 (GN–CU), but it increased in the same way from GN–CE to GN–CU. Raman shifts of G bands decreased in all samples below 1600 cm^−1^ (1592 cm^−1^, 1597 cm^−1^, 1598 cm^−1^), being the lowest in GN–CE and the highest in GN–CU. This can be explained by the layer-stacking enhancement in the matrix’s presence. Thus, the best filler dispersion was observed in GN–CU. Contrarily, the D-bands have been shifted towards higher Rs (1349 cm^−1^ in modified materials and 1350 cm^−1^ in GN). The D (first–order peak) moves towards lower frequencies in the case of tensile stress and towards higher ones in the presence of compressive stress, as observed in nanocomposites. Additionally, the I_D_ to I_G_ ratio decreases, especially in the case of the GN–CU, indicating better structural ordering. However, the sheer increase in the intensity of the PVC-derived band (2916 cm^−1^) in the materials after its functionalization may indicate better dispersion of these materials and deposition inside the polymer matrix [93] (the spectrum is collected from the polymer, not from the agglomerate on the surface of the thin polymer film).

AFM also investigated the surface structure of the obtained nanocomposites in contact with a surface (the upper side of the surface that was not in contact with glass during THF evaporation, forming a polymer film). AFM allows for the direct evaluation of the dispersion of graphene in the PVC matrix and the determination of roughness [94,96]. Figure 9 shows 2D and 3D images of nanocomposites containing modified and unmodified graphene.

The image of the surface topography of the nanocomposites shows some corrugation of the samples due to the evaporation of solvent during the formation of the polymer film [22]. AFM PVC/1%GN shows extreme inhomogeneity of the material, and the 3D image shows a large number of pointed tops, resulting from the presence of agglomerates. Additionally, the 2D image shows a lack of homogeneity of the material; the filler is distributed point–wise, and the sample is flat outside the place of its presence. In PVC/1%GN-CU nanocomposites, an increase in the homogeneity of the material was observed (no pointed vertices, no significant differences in roughness). At the same time, the roughness of the whole material increased significantly, which is a normal phenomenon in materials containing large amounts of dispersed filler [15,94,97]. However, the surface topography analysis showed the best dispersion in PVC films containing GN–CE; its material is homogeneous, and at the same time, its roughness is reduced compared to PVC/1%GN–CU.

The observations made earlier were confirmed by analyzing the homogeneity of the polymer films on a macroscopic scale. The presented digital images of nanocomposites containing 1 wt.% GN, GN–CU, and GN–CE Figure 10A–C, respectively, showed that the obtained materials had no visual defects in terms of their structure or the presence of pores. At the same time, it can be noted that the composite with unmodified GN is heterogeneous, and the filler is present in the form of agglomerates. Modifying the GN’s surface with curcuminoids significantly enhances the homogeneity of the nanocomposites, where the PVC/1% GN–CE material is characterized by the best graphene dispersion (the two phases in the material cannot be distinguished).

The same observations were confirmed by SEM imaging on the surface of the PVC/1%GN cryogenic breakthrough (Figure 10(A1)); we observed two phases in the material, and a graphene agglomerate is surrounded by a polymer matrix. A significant improvement in GN dispersion characterizes the PVC/1%GN–CU nanocomposites, but areas without filler can be observed on the surface of the cryogenic breakthrough (Figure 10(B1)). The image of the PVC/1% GN–CE breakthrough (Figure 10(C1)) is homogeneous throughout, and no areas of unmodified PVC can be seen, indicating the best filler dispersion [16,17,21,24]. At the same time, we can see that the structure of the nanocomposite differs significantly from that typically observed in unmodified poly(vinyl chloride) [20,21,22]. Its image may resemble balls of wool (Figure 10(C2)). Further zooming in on the structure (Figure 10(C3,C4)) shows the polymer matrix’s thorough coverage of individual graphene flakes, which may translate into a good filler–polymer interfacial interaction. At the same time, we can see that the modifier in the form of CE remains on the graphene flakes and does not pass into the PVC in significant amounts. The Figure 10(C4) image shows curcuminoid particles having spherical shapes [81,82] on the surface of the flakes. These spheres were not observed in the expanded ends of the polymer. The jaggedness of the PVC/GN–CE confirms that good filler dispersion further affects the fracture ductility of modified PVC.

Observations of the structure demonstrated the best improvement in the dispersion of graphene in materials modified by *Curcuma longa* L. extract. Therefore, the CE-modified filler was used to study the effect of the applied modification on the properties of poly(vinyl chloride)/graphene nanocomposites. Materials containing 0.01, 0.1, 0.5, and 1 wt% GN were produced.

### 2.8. Thermal and Mechanical Properties of Nanocomposites

Thermal properties were determined by the thermogravimetric method and the Congo Red test, with which the thermal stability time was measured. The results obtained are summarized in Table 2 and are shown in Figure 11. The TGA thermograms (Figure 11A) show mass losses at temperatures up to 170 °C, which are related to the evaporation of residual THF [20,23]. The content of solvent was about 6% (Table 2), despite the applied evaporation and drying methods (reduced pressure at 50 °C for 2 weeks). Earlier studies have shown that higher evaporation temperature would lead to the degradation of the material [20].

Thermal degradation of PVC and PVC/GN nanocomposites takes place in two stages related to the decomposition of the polymer. The first step occurs at 200 °C to 375 °C, and is related to the dehydrochlorination of PVC and the formation of conjugated polyene structures. The second step in the range from 375 °C to 600 °C corresponds to thermal cracking of the carbonaceous, conjugated polyene sequences and the formation of residual chars [11,23].

Due to the presence of residual THF in PVC, the nanocomposite films’ thermal stability was evaluated based on the differential thermogravimetry (DTG) [13,19]. DTG maxima of the first (max. DTG I) and second (max. DTG II) stages of decomposition, as well as the residual mass after the heating process, were used to analyze the thermal stability of the PVC/GN nanocomposites. The examples of thermogravimetric and DTG curves are presented in Figure 11A,B; all results are summarized in Table 2 (standard deviation of the obtained mean values in brackets). The max. DTG I analysis (Figure 11C) showed an increase in the thermal stability of graphene–CE modified materials, where the maximum temperature observed in the case of PVC/0.5%GN–CE was 8.5 °C higher compared to the unmodified matrix material. The introduction of GN–CE at a higher concentration does not cause a statistically significant change in thermal stability. In the case of nanocomposites with unmodified graphene, a significant improvement in stability was observed for materials with 0.1% GN; the temperature of the maximal degradation rate was in this case 3.2 °C higher than for poly(vinyl chloride). PVC/0.5%GN showed no statistically significant increase in this temperature, while an increase in standard deviation was observed. A further increase in the concentration of graphene filler to 1% had the effect of lowering the thermal stability of the nanocomposite materials. It can be assumed that the increase in standard deviation and decrease in thermal stability is due to the agglomeration of nanoparticles in the matrix material.

The max. DTG II analysis (Table 2) showed no statistically significant change in the temperature of the maximal degradation rate for materials with GN–CE. In contrast, a decrease in thermal stability was observed for materials with the unmodified graphene for concentrations of 0.5% and 1%, which may be related to the non-uniform distribution of the filler in the matrix. Residual mass analysis (Figure 11D) showed increases of these values in graphene–CE modified materials, and no statistically significant change in nanocomposites with GN. The increase in residual mass in materials with GN–CE indicates that a well-dispersed filler can act protectively against the carbon chain of the polymer, by forming an insulating layer [24,98].

A commonly used method for assessing the thermal stability of PVC materials is the Congo Red test. The results presented in Table 2 show that the introduction of graphene into polymer matrix results accelerates the dehydrochlorination process, which can take place according to various mechanisms [77]. The authors [19] indicated that the lower stability of PVC/GN composites may be due to the fact that GN nano-flakes act as reinforcing particulate fillers that attract chloride from the PVC. Obtained nanocomposites are sensitive to heat so they can undergo thermomechanical degradation processes under processing. Therefore, the use of thermal stabilizers will be required in the manufacture of these materials by conventional processing methods.

At the same time, it should be emphasized that discrepancies in the results of thermal stability measured by two methods are not unusual, due to the conditions under which the tests were conducted. Thermal stability by the Congo Red method is determined at a constant temperature that is close to the upper temperature range of PVC processing. In the TGA method, the sample is heated under a nitrogen atmosphere at a specified heating rate, over a wide temperature range, usually up to about 900 °C.

Figure 12 shows exemplary stress–strain curves for PVC with the nanocomposites on its matrix, and the determined tensile strength (TS) vs. GN and GN–CE concentration. The course of the stress–strain curves shows no yield point regardless of the material composition, which is as expected. The PVC/GN–CE nanocomposites had, compared to PVC/GN materials, a slightly higher mean TS value and a lower standard deviation, which is due to the better dispersion of the modified filler in the PVC matrix. However, the introduction of graphene into the PVC matrix, regardless of the graphene modification, did not resulted in a statistically significant change in the tensile strength of PVC nanocomposites.

An improvement of the mechanical properties of PVC/GN nanocomposites depends on the homogeneous dispersion of graphene in the polymer matrix and the structure of the filler. The lack of defects on the GN surface results in its high integrity and good mechanical properties, which directly determines the improvement of the mechanical properties of nanocomposites [26,99,100]. On the other hand, the dense GN network limits the penetration of poly(vinyl chloride) macromolecules between its layers, which makes the matrix discontinuous [14,24,100]. This explains the lack of a significant improvement in the mechanical properties of materials with GN–CE, where filler dispersion was improved.

### 2.9. Electrical Properties of Nanocomposites

Figure 13 shows the results of the resistivity tests of the obtained materials. The surface resistivity of poly(vinyl chloride) was determined to be 4.7 × 10^14^ Ω, and significant change occurs when the GN concentration in the composites is 0.5% and 1%. The lowest resistivity, which equals 1.4 × 10^7^ Ω, is characteristic of the PVC/1%GN sample, which makes it possible to conclude that it is an antistatic material [25,101]. Nanocomposites containing GN–CE did not show a decrease in surface resistivity, and the lowest value (1.5 × 10^14^ Ω) was shown by the sample PVC/1%GN–CE.

The volume resistivity of PVC/GN composites also decreases with the increase in graphene concentration in the matrix. The materials containing 1% of GN were characterized by the lowest resistivity, and equaled 9.8 × 10^5^ Ωm. Such a large change in resistivity (for PVC was 1.7 × 10^14^ Ωm) indicates that the dispersion of the GN was sufficient to create conduction paths. The PVC/GN–CE composites, as in the case of surface resistivity, did not show large changes in volume resistivity as the filler content in the matrix increased.

References [14,101] present that the influence of GN on the resistivity or conductivity of PVC/GN nanocomposites is significantly related to the dispersion of the filler in the polymer’s matrix. Although the modification of graphene with turmeric significantly improved the homogeneity of the structure of PVC/GN–CE nanocomposites, the expected improvement in electrical properties was not found. Graphene surface modification with curcuma extract is possible by π–π interactions. Some studies have shown the disturbances in the displacement of π electrons on the graphene’s surface, leading to the significant deterioration of its electrical properties [20,29]. This explains the lack of change regarding the resistivity of GN–CE-containing materials, despite a significant improvement of the graphene’s dispersion in the PVC matrix.

### 2.10. Swelling Behaviour of Nanocomposites

The numerous applications of poly(vinyl chloride) are due to its resistance to solvents [19,102,103]. However, PVC dissolves completely in THF and cyclohexanone, and it undergoes limited swelling when contacted with acetone. In this work, the chemical resistance of the proposed PVC/GN nanocomposites was tested by analyzing the swelling process in acetone. Swelling curves, i.e., the dependence of the degree of swelling as a function of time, are shown in Figure 14.

It was found that all the obtained materials undergo limited swelling, while the dependence of swelling degree on the exposure time to the swelling agent has a shape of a sigmoid function. Therefore, Equation (1) was used to approximate the swelling curves [20,22].
(1)$$S_{d}=\frac{S_{E}}{1+10^{\left(t_{M}-t\right)p}}$$
where:*S_d_* is swelling degree, %,*S_E_* is equilibrium swelling, upper asymptote, %,*t_M_* is time in which the swelling occurs with a maximum rate, s,*t* is time of exposure to the swelling agent, s,*p* is comparison parameter, 1 s^−1^.

The parameters of the equation and the coefficient of determination *R*^2^ are summarized in Table 3. The proposed model describes the experimental results with high accuracy, as evidenced by the high values of the coefficient of determination.

PVC/GN nanocomposites were found to have a lower equilibrium swelling ratio compared to unmodified PVC; the lowest SE value observed for PVC/1%GN, i.e., 43.5%, shows an improvement of about 5% in terms of the chemical resistance on the polymer matrix. Materials obtained with the addition of extract-modified graphene were characterized by a lower value of SE compared to corresponding materials with unmodified graphene. This effect was attributed to the better dispersion of the filler in the polymer matrix, as evidenced by significantly lower SE values for materials with 0.5 and 1% GN–CE. They were 32.1 and 27.3%, respectively, which represents a reduction in the equilibrium degree of swelling relative to PVC of 14.1%, in the case of PVC/0.5%GN–CE, and of 20.9%, in PVC/1%GN–CE. In addition, it should be noted that the agglomeration of the filler in the polymer matrix can lead to lower resistance to solvents in the form of acetone, which was observed in PVC/0.5%GN, where the SE was 1.3% higher relative to PVC. As the GN–CE content of the nanocomposites increases, the time in which the swelling occurs with a maximum rate (t_M_) also increases, which further confirms the correlation between acetone resistance of the materials and filler surface modifications. In materials with unmodified GN, a decrease in t_M_ values was observed with an increase in its content in the PVC matrix.

An increase in the chemical resistance of PVC, by using carbon fillers, occurs as a result of a decrease in its free volume. Consequently, this leads to a reduced access of solvents to the polymer chain [22,103]. In the case of the materials in question, this mechanism is very likely, as confirmed by the higher resistance of nanocomposites containing GN–CE, characterized by a better dispersion of graphene in the polymer matrix.

## 3. Materials and Methods

### 3.1. Materials

Graphene–based nanopowder with a flake thickness of 1.6 nm (maximum of 3 atomic monolayers), a flake length of 10 µm, and a specific surface area of 400 ÷ 800 m^2^g^−1^ was purchased from USA Graphene Laboratories Inc. Graphene was dispersed in a poly(vinyl chloride) solution using unmodified suspensive poly(vinyl chloride) Neralit 601 (Czech Republic, Spolana s.r.o. Anwil S.A. group), with a K number of 59–61, a bulk density of 0.56–0.63 g cm^−3^, a specific density of 1.39 g cm^−3^, and 97% purity. As a solvent for PVC and an environment used for graphene modification, tetrahydrofuran (THF) (Chempur, Piekary Śląskie, Poland) was used. Curcumin from *Curcuma longa* L. turmeric powder (Sigma-Aldrich, St. Louis, MO, USA) and *Curcuma longa* L. rhizome extract (made in-house [22]) was used to modify the surface of graphene. Methanol (Chempur, Piekary Śląskie, Poland) was used to wash modified graphene.

### 3.2. Graphene Modification

In order to functionalize the graphene’s surface with curcuminoids in the first step, 100 cm^3^ solutions of CU and CE in THF with a concentration of 2.25 mg cm^−3^ were prepared by ultrasound (frequency of 20 kHz, 40% amplitude, time 10 min, temperature 23 °C) using a SONOPULS rod-shaped probe homogenizer from Bandelin. Graphene was then introduced into the solution, which was prepared so that the dispersion concentration was 1.5 mg cm^−3^. Next, GN was sonicated under the same conditions as curcuminoids for 2 h. It was then stirred with a magnetic stirrer for 15 h at 200 rpm. After this time, the material was centrifuged 4 times at 12,000 rpm for 4 min and washed after each step with methanol, in order to remove the undeposited modifier. After centrifugation, the graphene was dried at 45 °C for 65 h to evaporate residual solvents.

### 3.3. Preparation of PVC/GN Dispersions and Nanocomposites

In the first step, poly(vinyl chloride) was dissolved in THF at 25 °C for about 48 h, yielding a solution of 3 wt%. Then, the dry graphene was added to the solution and dispersed using ultrasound (conditions as in Section 2.2) for 60 min. The amount of graphene in the prepared dispersions for dispersion stability testing was 1 wt% per polymer mass.

Thin films of PVC/GN, PVC/GN–CU, and PVC/GN–CE nanocomposites containing 1% filler for structure test were obtained by solvent evaporation, where the dispersions were poured onto Petri dishes with a diameter of 7 cm, and the solvent was evaporated at 50 °C for 24 h. Nanocomposites for testing properties were dried in a vacuum drier under a reduced pressure (max 20 mbar absolute) at 50 °C for 2 weeks, in order to remove the THF residue from the polymer films. All samples were coded to take into account the graphene content and the presence of CE, e.g., a sample containing GN at a concentration of 0.01 wt.% was coded as PVC/1%GN, whereas a sample containing the same graphene modified by *Curcuma longa* L. rhizome extract was coded as PVC/1%GN–CE, and that modified by Curcumin from *Curcuma longa* L. turmeric powder was consequently coded as PVC/1%GN–CU.

### 3.4. Characterization

Absorption studies were conducted using a UV-vis/NIR spectrometer (Perkin Elmer Lambda 1050+) in a quartz cuvette. For this purpose, solutions of CU, CE in methanol with a concentration of 0.008 mg cm^−3^ and dispersions of GN, GN–CU, and GN–CE with a concentration of 0.03 mg cm^−3^ were prepared. The study was conducted in the 210–800 nm range. The experimental results were normalized in the range 0–1 using the OriginLab software.

X–ray photoelectron spectroscopy: a PHI 5000 VersaProbe (ULVAC–PHI Inc., Hagisono, Japan) spectrometer was used to conduct XPS measurements under the following conditions: monochromatic Al Kα radiation (hν = 1486.6 eV), an X-ray source operating at 25 W, 15 kV, 100 µm spot, pass energy of 23.5 eV, energy step of 0.1 eV. Obtained XPS spectra were analyzed with CasaXPS software, using the set of the sensitivity factors native for the hardware. Shirley background and Gaussian–Lorentzian peak shape were used for deconvolution of all spectra.

The thermal stability of the fillers and nanocomposites was assessed by the thermogravimetric method using a TG 209 F3 Tarsus apparatus (Netzsch). The heating rate was about 10 °C min^−1^ in an open ceramic crucible, under a nitrogen atmosphere, at 30 to 700 °C for fillers and at 30 to 900 °C for nanocomposites. In addition, static thermal stability tests were determined for the nanocomposite materials using the Congo Red test. The test was carried out at 200 °C, measuring the thermal stability time, i.e., the time during which the sample shows no signs of degradation in the form of hydrogen chloride release and color change of the indicator paper.

Raman spectra were collected with a Raman DXR microscope (Thermo Fisher Scientific, Waltham, MA, USA) using two laser lines, 532 nm and 780 nm, where the shorter wavelength was used for polymer composites, and fillers were studied using the longer wavelength line. The specimens were measured with 50 repetitions of 50 s each (20 in successive control measurements). The aperture was 50 µm, the lenses were 10 and 50 mm, and the laser beam power was 0.1 mW for 780 nm and 1 mW for 532 nm. Each sample was measured at least dozens of times (from 80 to >300) at different locations, and the final signal was obtained as an average and statistically representative one. The morphology of the samples was studied using scanning electron microscopy (SEM), and a Zeiss Crossbeam 350 microscope; in the case of polymer nanocomposites, cryogenic breakthroughs with a sputtered gold layer were observed. As a second technique, a Zeiss Libra 120 Plus transmission electron microscope (TEM) (Stuttgart, Germany) operating at 120 kV was used, where the aqueous suspension was placed on a copper grid coated with a formvar polymer layer. Finally, the sample was left in the air (under a fume hood) to dry.

Stability testing of the obtained graphene dispersions in PVC solution was carried out by multiple light scattering using the Turbiscan Lab apparatus (Formulaction SA, Toulouse, France), where suspensions of GN, GN–CU and GN–CE in 3% PVC solution in THF were placed in glass cylindrical test tubes, with a working height of 54 mm. The samples were then placed in a Turbiscan Lab instrument and scanned with light at approximately 880 nm. Scanning was carried out every 1 h for 24 h at room temperature (23 °C). The Turbiscan Stability Index (TSI) was determined from the results.

Filler dispersion in the matrix was also determined using an atomic force microscope Nanosurf (Liestal, Switzerland) in contact mode with the PPP–XYCONTR probe (NANOSENSORS, Neuchatel, Switzerland), where the radius of the aluminum-coated blade was 7 nm, while the line angle was 20 degrees. 2D and 3D images were taken within a 50 × 50 µm area. To determine the surface topography of the area, 500 lines were measured (at 20 nm intervals), and the number of measurement points per line was 5000. The scanning speed of the sample was 5 µm s^−1^.

A study of tensile properties was carried out using an in-house-built tensile tester, with the parameters presented in our earlier work [22]. The tests were carried out on samples with a thickness of 0.2 ± 0.07 mm and a width of 2 ± 0.1 mm, while the length of the measuring section L0 was 1.5 mm. The materials were stretched at a constant speed of 0.1 mm s^−1^.

The electrical properties of nanocomposites were determined by measuring system consisting of a 6517A electrometer and a 8009 measuring chamber (Keithley Instrument Inc., Cleveland, OH, USA). The surface and volume resistivity were measured. The tests were carried out on film samples with a diameter of 70 mm in air, at a temperature of 23 °C and humidity of 50%, and at a voltage of 10 V.

The resistance of the obtained materials to swelling in acetone was tested in accordance with the method proposed in our earlier study [20]. The change in the swelling degree (*S_d_*) was determined (see Equation (2)), depending on the immersion time in the swelling agent. The changes in the samples’ diameters were set on the basis of photos using the NIS Elements 4.0 software. The frequency picture-taking depended on the exposure time to the swelling agent. The measurement temperature was 20 °C and the initial diameter of the samples *h*_0_ = 10 mm.
(2)$$S_{d}=\frac{h-h_{0}}{h_{0}}\times 100\%$$
where:*h* is sample diameter after time *t* (mm),*h*_0_ is initial sample diameter (mm).

In order to analyse the obtained results, Origin 8.6 Pro software with implemented statistical analysis modules was used. ANOVA with Tukey’s post–hoc test was used to compare the significance of the difference for the mean values of the obtained results. The normal distribution was confirmed by the Shapiro–Wilk test, and the homogeneity of variance by the Levene’s test. All analyses were performed assuming a significance level below 0.05.

## 4. Conclusions

Graphene is one of the most promising nanofillers of nanocomposites, improving their performance. However, its uniform distribution in the polymer matrix is challenging. Several studies refer the need to reduce the agglomeration of GN and its derivatives during nanocomposite fabrication, in order to achieve improved properties [104,105,106,107]. In the present work, an environmentally safe method of graphene surface modification is presented, using curcumin and an extract derived from the powdered rhizome of *Curcuma longa* L.

Observations of the modified materials’ morphology showed better CE dispersion on the graphene’s surface. Spectroscopic studies have shown that modification occurs via a π–π stacking mechanism. TGA analysis revealed a higher amount of deposited curcumin on GN flakes i.e., 11.4 wt.%, while for the extract it was 6.9 wt.%. In addition, it is worth noting that the modified materials have a high thermal stability of about 340 °C. The analysis of the stability of GN, GN–CU, and GN–CE dispersions, as well as observations of the polymer films structure obtained after solvent evaporation from the dispersions, showed a significant improvement in the homogeneity of the nanocomposites after modification by curcuminoids. At the same time, extract-modified graphene showed better dispersion improvement in PVC. That indicates that not only the amount of modifier deposited on the graphene surface, but also its size and dispersion important in improving PVC/GN homogeneity. The analysis of the properties of nanocomposites with GN-CE has shown a better thermal and chemical stability of these materials. On the other hand, based on the results of the Congo Red test, it was stated that the proposed nanocomposites will require thermal stabilization at the processing stage by traditional methods, which is a well-known phenomenon in the field of the PVC technology.

Summarizing, the proposed naturally derived compounds in the form of CE can be an alternative to previously proposed modifications of graphene for use in PVC and other polymer composites. In addition, the high thermal stability of the proposed modification, as well as the relatively low cost and possibility of obtaining the modifier on a large scale, provide opportunities for the technological use of CE in the future production of graphene polymer nanocomposites, through traditional processing methods at high temperatures.

## Figures and Tables

**Figure 1 molecules-28-03383-f001:**
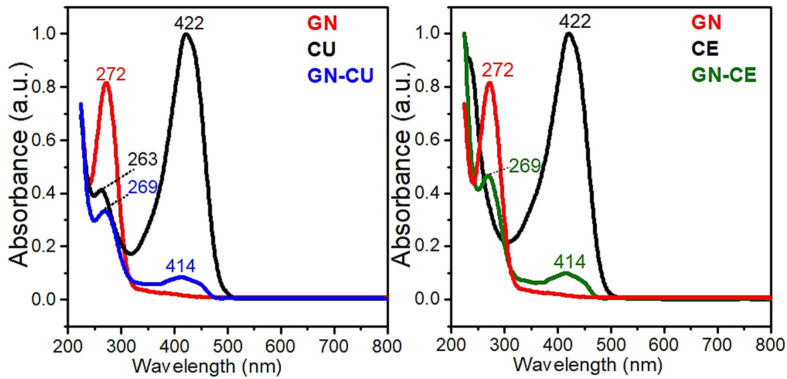
UV–Vis spectra of the GN, CE, CU and GN–CU, GN–CE dispersion in methanol.

**Figure 2 molecules-28-03383-f002:**
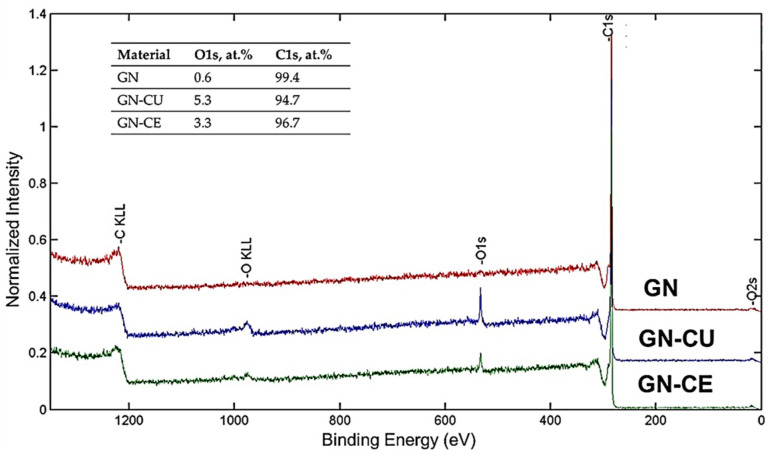
XPS spectra of the GN, GN–CU and GN–CE.

**Figure 3 molecules-28-03383-f003:**
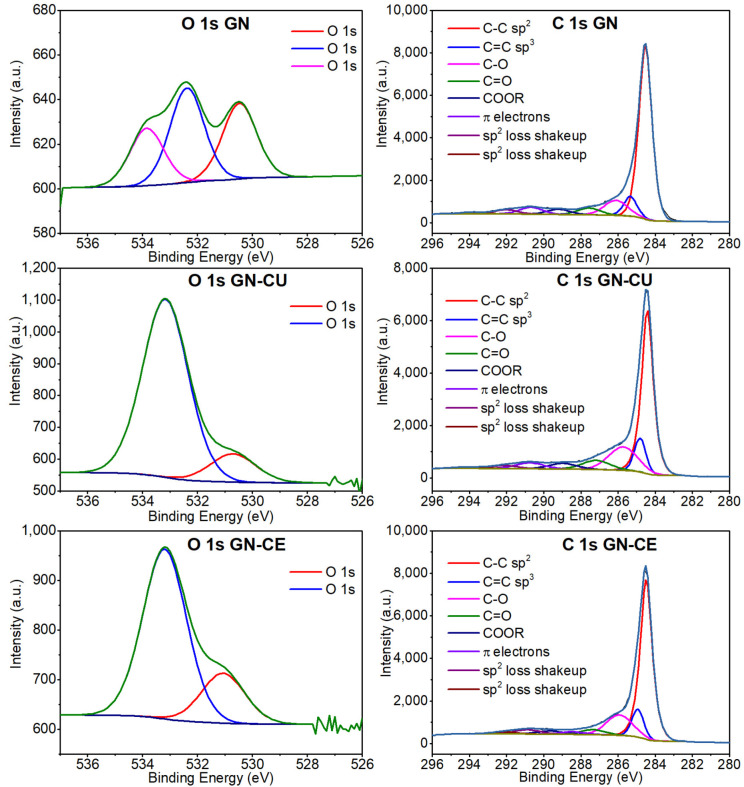
Peaks deconvolution of O1s and C1s of the GN, GN–CU and GN–CE.

**Figure 4 molecules-28-03383-f004:**
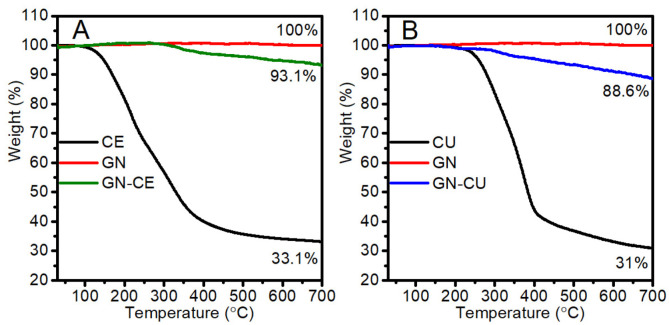
TGA of curcumin, GN, GN–CU (**A**), and *Curcuma longa* L extract and GN, GN–CE (**B**).

**Figure 5 molecules-28-03383-f005:**
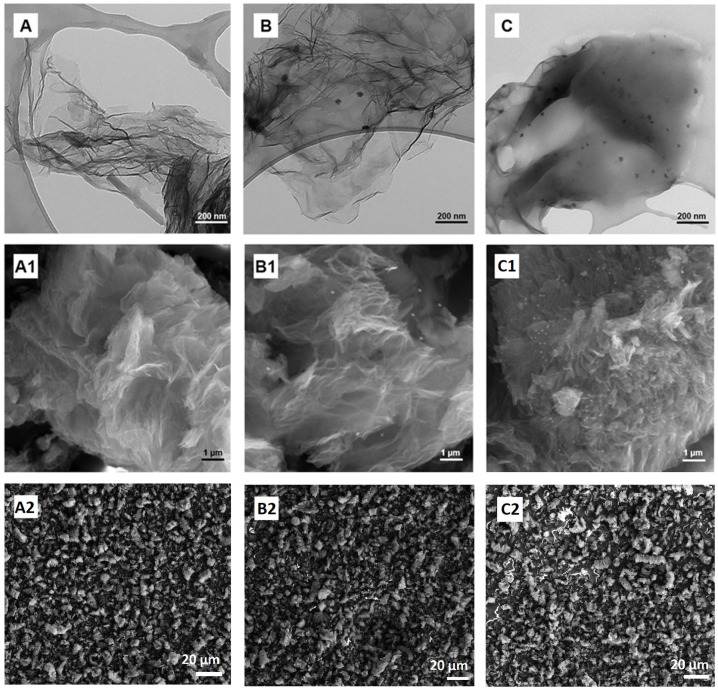
TEM images of GN, GN–CU, GN–CE (**A**–**C**), SEM images of GN, GN–CU, GN–CE (**A1**,**B1**,**C1**) with scale bar 1 µm, and GN, GN–CU, GN–CE (**A2**,**B2**,**C2**) with scale bar 20 µm (survey images).

**Figure 6 molecules-28-03383-f006:**
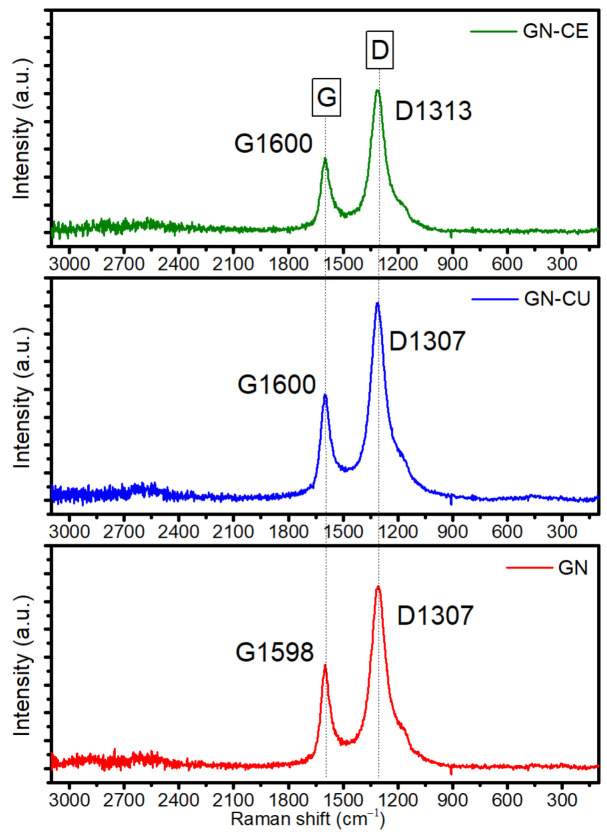
Raman spectra of GN, GN–CU and GN–CE.

**Figure 7 molecules-28-03383-f007:**
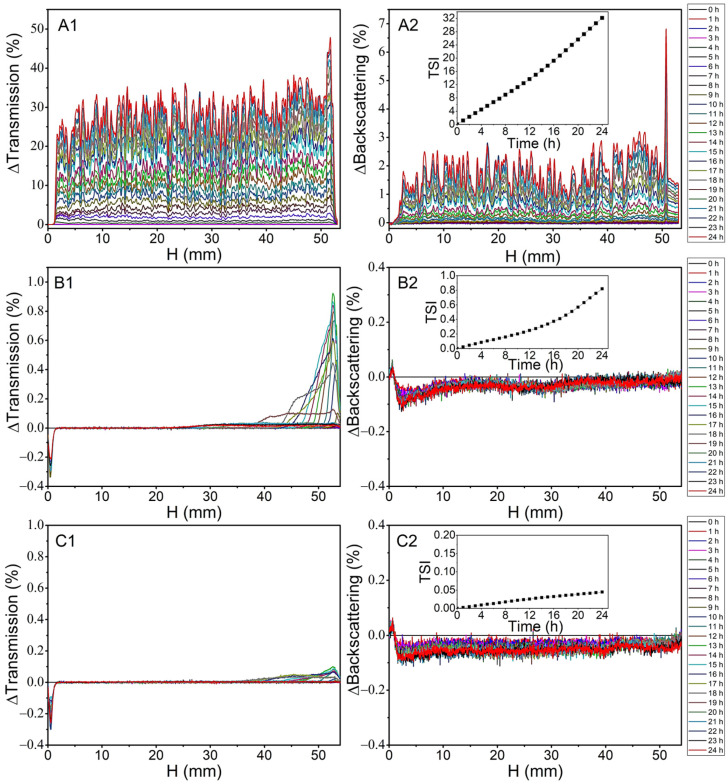
ΔTransmission and ΔBackscattering of GN (**A1**,**A2**), GN–CU (**B1**,**B2**) and GN–CE (**C1**,**C2**) dispersion.

**Figure 8 molecules-28-03383-f008:**
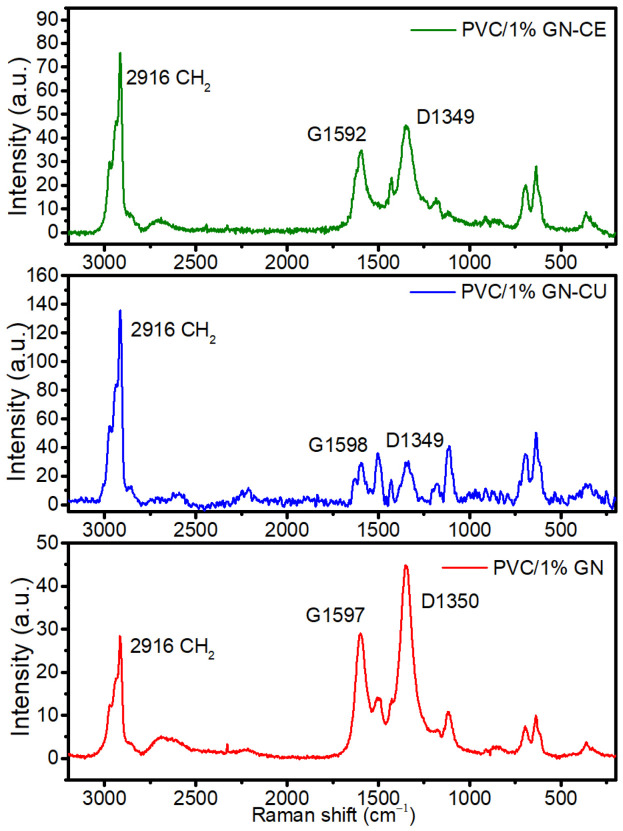
Raman spectra of PVC/1%GN, PVC/1%GN–CU and PVC/1%GN–CE.

**Figure 9 molecules-28-03383-f009:**
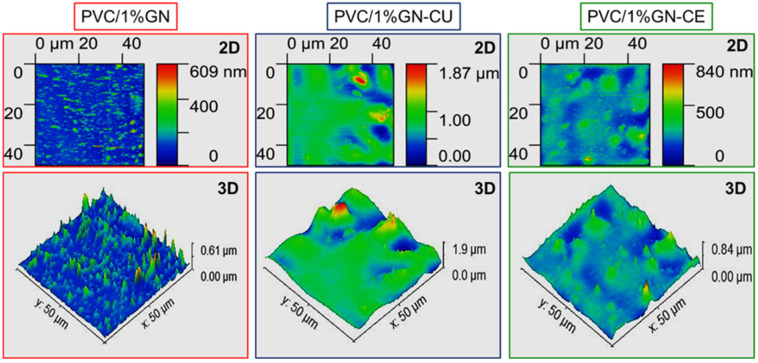
AFM 2D and 3D images of PVC/1% GN, PVC/1%GN–CU and PVC/1% GN–CE.

**Figure 10 molecules-28-03383-f010:**
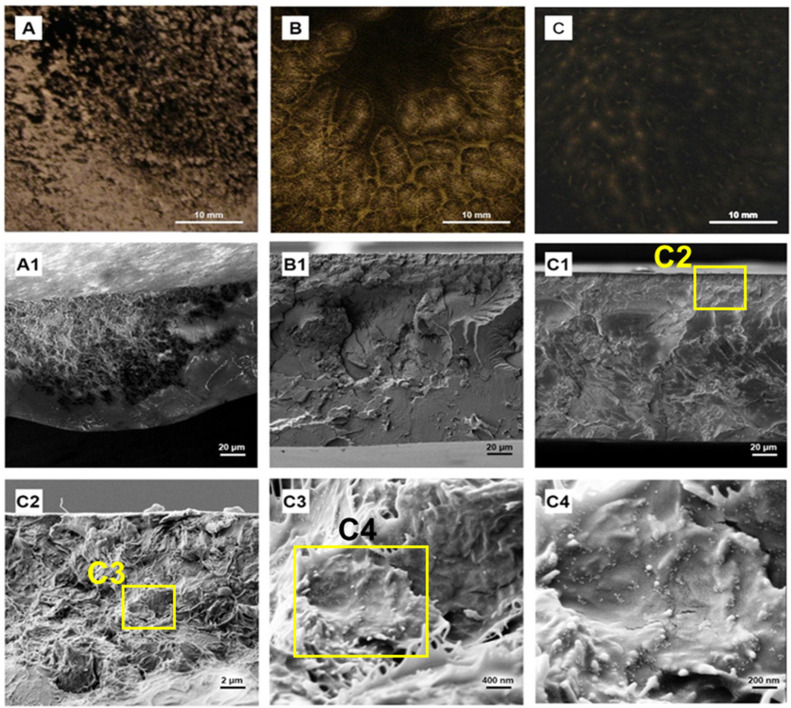
Digital photos of PVC/1% GN, PVC/1%GN–CU and PVC/1% GN–CE (**A**–**C**), SEM images of those materials (**A1**,**B1**,**C1**) and zoomed in structure of PVC/1%GN–CE (**C2**–**C4**).

**Figure 11 molecules-28-03383-f011:**
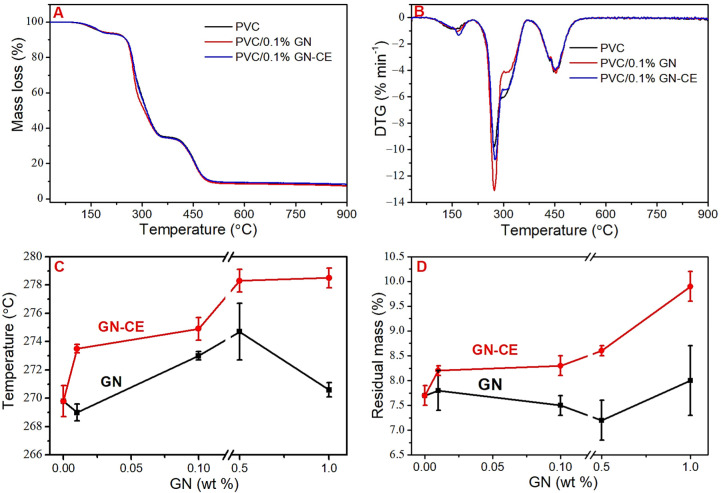
(**A**) Example TGA thermograms, (**B**) Example DTG curves, (**C**) Max. DTG I analysis, (**D**) Residual mass analysis.

**Figure 12 molecules-28-03383-f012:**
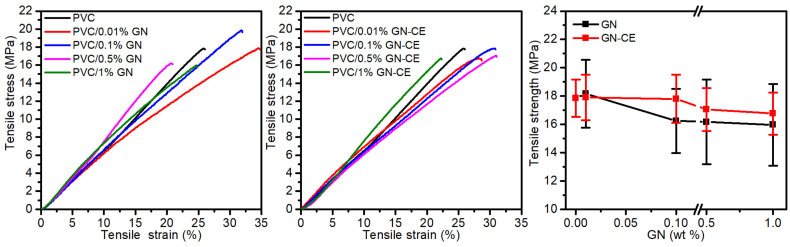
Mechanical properties of PVC, PVC/GN and PVC/GN-CE nanocomposites.

**Figure 13 molecules-28-03383-f013:**
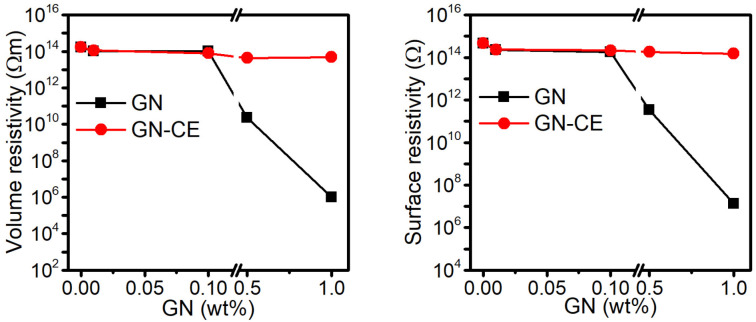
Surface and volume resistivity of PVC, PVC/GN and PVC/GN–CE nanocomposites.

**Figure 14 molecules-28-03383-f014:**
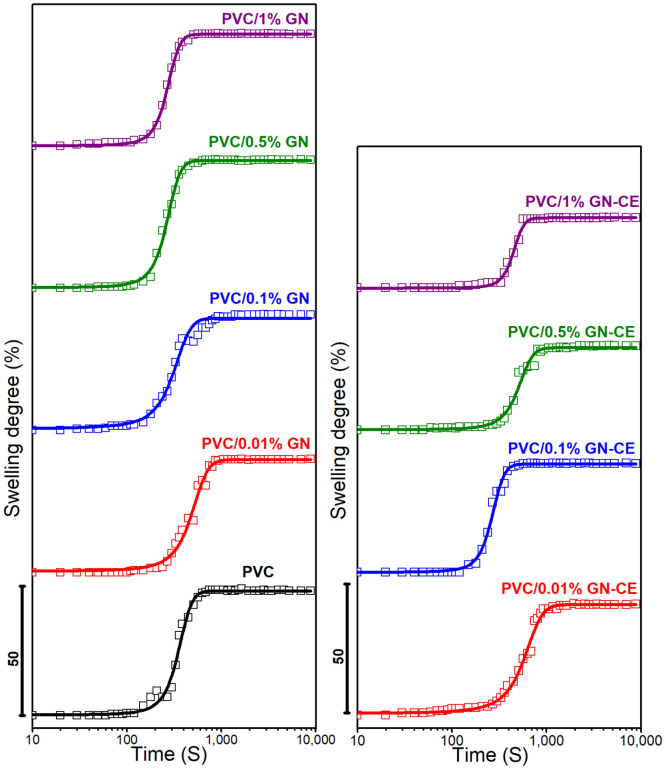
Swelling degree of PVC, PVC/GN and PVC/GN–CE nanocomposites vs. time.

**Table 1 molecules-28-03383-t001:** XPS characterization of GN, GN–CU and GN–CE.

Binding Energy, eV	Chemical Bonds	FWHM	Atomic, %
GRAPHENE			
O1s			
530.47	O1s	1.47	0.2
532.38	O1s	1.47	0.2
533.84	O1s	1.47	0.2
			**0.6**
C1s			
284.53	C-C sp^2^	0.82	65.3
285.33	C=C sp^3^	0.82	7.5
286.11	C-O	1.49	10.0
287.56	C=O	1.49	4.5
289.21	COOR	1.49	3.5
290.68	π electrons	1.49	4.6
292.03	sp^2^ loss shakeup	1.49	2.8
293.87	sp^2^ loss shakeup	1.49	1.1
			**99.4**
			**100.0**
GRAPHENE–CURCUMIN			
O1s			
530.7	O1s	1.95	0.7
533.17	O1s	1.95	4.6
			**5.3**
C1s			
284.42	C-C sp^2^	0.73	50.9
284.81	C=C sp^3^	0.73	9.6
285.72	C-O	1.82	16.5
287.18	C=O	1.82	6.4
288.99	COOR	1.82	4.0
290.73	π electrons	1.82	4.2
292.14	sp^2^ loss shakeup	1.82	2.1
294.15	sp^2^ loss shakeup	1.82	1.0
			**94.7**
			**100**
GRAPHENE–*Curcuma Longa* L. EXTRACT			
O1s			
531.06	O1s	1.87	0.8
533.19	O1s	1.87	2.5
			**3.3**
C1s			
284.48	C-C sp^2^	0.76	59.4
284.94	C=C sp^3^	0.76	9.4
285.92	C-O	1.65	15.2
287.35	C=O	1.65	3.7
289.66	COOR	1.65	2.4
288.45	π electrons	1.65	2.0
290.88	sp^2^ loss shakeup	1.65	3.2
292.16	sp^2^ loss shakeup	1.65	1.5
			**96.7**
			**100.0**

**Table 2 molecules-28-03383-t002:** Thermal properties of PVC/GN and PVC/GN–CE nanocomposites.

Material	Cont. of THF, %	Max. DTG I, °C	Max. DTG II, °C	Residual Mass, %	Congo Red Test, min
PVC	5.2 (0.4)	269.8 (2.1)	452.4 (1.5)	7.7 (0.2)	3.5 (0.02)
PVC/0.01%GN	5.8 (0.1)	269.0 (0.6)	450.9 (0.6)	7.8 (0.4)	3.1 (0.04)
PVC/0.01%GN–CE	5.5 (0.1)	273.5 (0.3)	449.9 (1.1)	8.2 (0.1)	3.3 (0.08)
PVC/0.1%GN	5.8 (0.4)	273.0 (0.3)	454.2 (0.4)	7.5 (0.2)	3.1 (0.04)
PVC/0.1%GN–CE	6.4 (0.1)	274.9 (0.8)	452.9 (1.4)	8.3 (0.2)	2.8 (0.03)
PVC/0.5%GN	5.7 (0.6)	274.7 (2.0)	446.8 (0.6)	7.2 (0.4)	2.8 (0.04)
PVC/0.5%GN–CE	6.1 (0.2)	278.3 (0.8)	451.4 (1.5)	8.6 (0.1)	2.4 (0.03)
PVC/1%GN	6.8 (0.5)	270.6 (0.5)	438.1 (2.9)	8.0 (0.7)	2.6 (0.08)
PVC/1%GN–CE	6.6 (0.2)	278.5 (0.7)	453.5 (1.3)	9.9 (0.3)	2.3 (0.03)

**Table 3 molecules-28-03383-t003:** Parameters of the model describing the swelling process.

Material	*S_E_*, %	*t_M_*, s	*p,* s^−1^	*R* ^2^
PVC	48.2 (0.3)	348 (4)	0.007 (0.0004)	0.994
PVC/0.01% GN	43.8 (0.4)	477 (7)	0.004 (0.0002)	0.994
PVC/0.01% GN–CE	42.9 (0.3)	570 (7)	0.003 (0.0001)	0.996
PVC/0.1% GN	43.9 (0.4)	307 (6)	0.006 (0.0004)	0.990
PVC/0.1% GN–CE	42.1 (0.2)	262 (3)	0.010 (0.0005)	0.996
PVC/0.5% GN	49.5 (0.2)	262 (2)	0.009 (0.0003)	0.998
PVC/0.5% GN–CE	32.1 (0.3)	498 (8)	0.004 (0.0003)	0.992
PVC/1% GN	43.5 (0.1)	267 (1)	0.010 (0.0002)	0.999
PVC/1% GN–CE	27.3 (0.2)	443 (4)	0.007 (0.0003)	0.997

## Data Availability

The data that support the findings of this study are available from the corresponding author upon reasonable request.

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
