# Peer review of "Graphene Modification by Curcuminoids as an Effective Method to Improve the Dispersion and Stability of PVC/Graphene Nanocomposites"

_molecules, 2023, doi:10.3390/molecules28083383_

Round 1

Reviewer 1 Report

The paper “Graphene modification by curcuminoids as an effective method 2 to improve the dispersion and stability of PVC/graphene nano-composites.

Is innovative work presenting an effective method to improve the diffusion of graphene in PVC solution

From experimental point of view, is interesting and can be accepted for publication provide the author comply the above the following minor revisions:

-        Please write better the abstract lines 27,28,29

-        Please in the line 35 indicate GN graphene nanocomposite or remove N if is referred only to graphene(G)

-        Regarding the XPS analyses could be more quantitative present a ratio of the O1s signal respect C1s one

-        Fig3 is it possible to have more integrate O1s signal? Or is it possible an post process average of the signal, in order to reduce a little bit the signal to noise related to the less O1s intensity?

-        Fig 4 please insert Panel A and Panel B in the picture and in the caption and try to explain better the difference between two panels

-        Please modify figure 6 with line indicating the shift

-        Please insert an insert with the zoom of Fig C1

-        Please adjust table 3 format

-        Please adjust fig 14

-        For the XPS analyses did you use an neutralizator system?wich kind of neutralizator system did the author use?

-        the paper has a very extensive bibliography, is it possible to add more recent ones?

Author Response

Dear reviewer,

 Thank you very much for reviewing our work and the valuable comments that helped to improve it. Please see below for detailed responses.

  • “Please write better the abstract lines 27,28,29” Abstract lines 27,28,29 have been corrected.
  • “Please in the line 35 indicate GN graphene nanocomposite or remove N if is referred only to graphene(G)” The abbreviation we used came from Graphene nanoplates, the text in line 35 has been corrected so that it is not misleading.
  • “Regarding the XPS analyses could be more quantitative present a ratio of the O1s signal respect C1s one” In fig 2. a table has been added showing the ratio of O1s signal to C1s.
  • “Fig3 is it possible to have more integrate O1s signal? Or is it possible an post process average of the signal, in order to reduce a little bit the signal to noise related to the less O1s intensity?” Figure 3 has been corrected
  • Fig 4 please insert Panel A and Panel B in the picture and in the caption and try to explain better the difference between two panels”. In Fig. 4, Panel A and Panel B are introduced, and the corresponding figure references are inserted in the text.
  • Please modify figure 6 with line indicating the shift” Figure 6 has been corrected
  • “Please insert an insert with the zoom of Fig C1” Changes have been made to fig. 10.
  • Please adjust table 3 format. Please adjust fig 14” Table 3 and Figure 14 have been corrected
  • “For the XPS analyses did you use an neutralizator system?wich kind of neutralizator system did the author use?” In measurements was used 2 methods of neutralizing the surface charge that Versa has:
    • voltage applied to the station/carrier/sample
    • bombardment with low-energy Ar+ ions using an ion cannon
  • “the paper has a very extensive bibliography, is it possible to add more recent ones?” Thank you for your valuable comment, which we agree with. In the bibliography where possible, 2 items from 2023 have been added. However, we would like to emphasize that despite the numerous studies on modification of polymers with graphene. The literature on poly(vinyl chloride) modification is not extensive, so the paper cites publications from the last 10 years, which account for more than 80% of the citations. Where possible, we used the latest literature on PVC/graphene composites.

Reviewer 2 Report

The paper deals with the directed modification of graphene with curcumin and related substances to render a more advanced compound for composites with polymers in making thin film. The selection of methods is quite correct as the the discussion and implementation of this characterization. I believe that the paper can be published upon correcting the following minor issues.

- Almost every figure have three lines, for graphene, modifier, and the composition, and usually th line colors are black, red, and blue (or green). But every time, color selection is different, so graphene may be red, black or blue, etc. In my opinion, for a reader, the same color for the same case is more appropriate. Please consider revising the figures.

- Some minor formatting errors are found throughout the text, unnecessary periods (like in the title), dashes instead of n-dash and minus signs, the use of dash instead of a comma or the word 'is' in the explanation of parameters in formulae, etc.

- I suggest expanding Conclusions a bit, as it now is a kind of summary, which repeats the Abstract and Discussion. Probably, more outlooks or expanding the findings of this research to other materials could be briefly outlined here.

Author Response

Dear Reviewer,

Thank you very much for reviewing our work and the valuable comments that helped to improve it. Please see below for detailed responses.

  • “Almost every figure have three lines, for graphene, modifier, and the composition, and usually th line colors are black, red, and blue (or green). But every time, color selection is different, so graphene may be red, black or blue, etc. In my opinion, for a reader, the same color for the same case is more appropriate. Please consider revising the figures. „

The colors in the charts have been standardized. As follows: red is for graphene, blue is for curcumin-modified graphene, green is for extract-modified graphene. If the properties of modifiers were analyzed, they are black. When comparing polymer nanocomposites, black color was used for materials with graphene and red for materials with extract-modified graphene.

  • “Some minor formatting errors are found throughout the text, unnecessary periods (like in the title), dashes instead of n-dash and minus signs, the use of dash instead of a comma or the word 'is' in the explanation of parameters in formulae, etc.”

Punctuation marks were corrected  marks throughout the text (without marking changes). We hope that all errors have been eliminated.

  • “I suggest expanding Conclusions a bit, as it now is a kind of summary, which repeats the Abstract and Discussion. Probably, more outlooks or expanding the findings of this research to other materials could be briefly outlined here.”

Conclusions have been corrected